# Population-based microbiological characterization of *Nocardia* strains causing invasive infections during a multiyear period in a large Canadian healthcare region

Deirdre Church,[1,2,3] Gisele Peirano,[1] Alejandra Ugarte-Torres,[1] Christopher Naugler[2,3,4]

**ABSTRACT** The role of specific *Nocardia* species in human infections continues to expand as advanced methods enable species/species complex identification. We conducted a multiyear population-based (2010–2022) characterization of invasive *Nocardia* spp. Isolates were analyzed using matrix-assisted laser desorption ionization-time of flight mass spectrometry (MALDI-TOF MS) and 16S rRNA gene sequencing. A total of 94 cases and 117 unique 16S rRNA gene sequences were evaluated from pulmonary (53%), skin and soft tissues (SSTIs) (29%), central nervous system (CNS) (7%), bloodstream (6%), and other sites of infection. Infections were mainly caused by three species complexes: *N. farcinica* (*n* = 20, 21.3%), *N. cyriacigeorgica* (*n* = 16, 17%), and *N. nova* (*n* = 15, 16%). Phylogenetic analysis correlated with the clinical site of infection. Most *N. cyriciageorgica* (92%) complex isolates caused SSTIs or pulmonary infections, and most *N. farcinica* complex (67%) and *N. nova* complex (69%) isolates caused pulmonary and CNS infections, respectively. Several other unique *Nocardia* spp. rarely caused invasive infections (≤5 cases): (i) pulmonary (*N. abscessus* complex, *N. amamiensis*, *N. asteroides*, *N. otitidiscaviarum* complex, *N. sputorum*, *N. transvalensis* complex, *Nocardiopsis* spp., *Pseudonocardia* spp.); (ii) SSTIs (*N. otitidiscaviarum* complex); and (iii) CNS (*N. flavorosea* and *N. paucivorans* complex). *Nocardia* species were highly susceptible to amikacin, trimethoprim-sulfamethoxazole, moxifloxacin, and linezolid. Imipenem resistance occurred in the *N. farcinica* complex and *N. nova* complex, while ceftriaxone resistance only occurred in the former. Antibiotic profiles varied for rare *Nocardia* spp. Species-level *Nocardia* identification using MALDI-TOF MS and 16S rRNA gene sequencing improves understanding of these organisms' unique roles in causing invasive disease.

**IMPORTANCE** *Nocardia* spp. are a rare cause of invasive infections, particularly in immunocompromised patients. The role of specific *Nocardia* species in human infections continues to expand as advanced methods enable species/species complex identification. We conducted a multiyear population-based (2010–2022) characterization of invasive *Nocardia* spp. Isolates were analyzed using matrix-assisted laser desorption ionization-time of flight mass spectrometry (MALDI-TOF MS) and 16S rRNA gene sequencing. A total of 94 cases included pulmonary infections (53%), skin and soft tissue infections (29%), central nervous system infections (7%), bloodstream infections (6%), and septic arthritis and intra-abdominal infections (5%). One hundred seventeen unique 16S rRNA gene sequences from clinical isolates were analyzed. Phylogenetic analysis correlated with the clinical site of infection. *Nocardia* species were highly susceptible to amikacin, trimethoprim-sulfamethoxazole, moxifloxacin, and linezolid. Imipenem resistance only occurred in *N. farcinica* complex and *N. nova* complex strains, and ceftriaxone resistance only occurred in the former. Species-level *Nocardia* identification

**Peer Reviewer** Benjamin Douglas Moser, Centers for Disease Control and Prevention, Atlanta, Georgia, USA

Address correspondence to Deirdre Church, dchurch@ucalgary.ca.

The authors declare no conflict of interest.

using MALDI-TOF MS and 16S sequencing improves understanding of these organisms' unique roles in invasive disease.

**KEYWORDS** *Nocardia*, 16S rRNA gene sequencing, analysis, invasive infection, skin and soft tissue infection, pulmonary infection, central nervous system infection, bloodstream infection, MALDI-TOF MS, phylogenetic analysis, antimicrobial susceptibility

Nocardiae are part of the aerobic actinomycetes belonging to the class *Actinobacteria* in the order *Corynebacteriales* along with other clinically important genera, including *Corynebacterium* and *Mycobacterium* (1, 2). They are opportunistic soil pathogens that are emerging as uncommon causes of invasive infections, primarily pulmonary, central nervous system (CNS), bloodstream (BSI), and skin and soft tissue infections (SSTIs) in immunocompromised patients (1–10). Recent widespread adoption of initially genomic methods (i.e., partial sequencing of the 16S rRNA gene), then proteomics methods (i.e., matrix-assisted laser desorption ionization-time of flight mass spectrometry [MALDI-TOF MS]) for routine use by clinical microbiology laboratories has recently allowed a more precise species-level characterization of this complex group of gram-positive organisms. Genomic analyses have also dramatically expanded the number of known *Nocardia* spp. from only a few in the 1990s to the currently 109 recognized species with valid names according to the List of Prokaryotic Names with Standing in Nomenclature (LPSN) (http://www.bacterio.net/index/html). About half of all known species, however, have been documented to be clinically significant based on published case reports in the literature (1). To date, *Nocardia asteroides* remains as the most reported species from human specimens (1), but few cases have been caused by this species in our region.

Our large centralized regional clinical laboratory has routinely used fast partial sequencing of the 16S rRNA gene (V1 to V3 region) to identify a wide variety of gram-positive bacilli for the past two decades (11) and implemented MALDI-TOF MS in 2018 for the routine initial identification of *Nocardia* spp. Although rare, increasing numbers of invasive infections have been diagnosed using these methodologies in our large Canadian healthcare region. We performed a multi-year isolate characterization and a population-based epidemiological analysis of the >90 cases diagnosed between 2010 and 2022 because few population-based studies have been globally reported. We also sought to define the specific *Nocardia* species/species complexes causing nocardiosis in our region, as there are no population-based reports from Canada. A phylogenetic analysis of all available 16S rRNA sequence data was used to determine if any *Nocardia* spp. were associated with specific types of infection and provide additional confirmation and clarity of discordant identifications between proteomics and genomic analyses. Antimicrobial susceptibility testing was also performed on all isolates to compare the antibiograms found locally with previously reported data.

## MATERIALS AND METHODS

### Study design

We completed a retrospective population-based cohort study at the centralized regional microbiology laboratory serving the Calgary Zone and surrounding areas from 1 January 2010 to 31 December 2022. The Calgary Laboratory Services (now Alberta Precision Laboratories, Alberta Health Services [AHS]) is a centralized regional microbiology laboratory providing ambulatory and hospital-based testing to Calgary and the South Zone (AHS), covering a population of ~2.6 million. Basic demographic data for invasive *Nocardia* spp. isolates, such as patient age and sex, infectious diagnosis, underlying comorbidities, and 1-year mortality, were determined through a retrospective chart review of most patients' electronic medical records, which was independently performed by two investigators (A.U.T. and D.L.C.). All clinical diagnoses and the site(s) of nocardiosis infection were determined as clinician-defined from the problem list by a detailed

clinical chart review. The site of infection was based on clinical symptoms and diagnostic imaging and laboratory evidence of nocardial disease. Invasive infection was defined as a laboratory-confirmed *Nocardia* spp. infection at a primary site, either pulmonary, skin, and soft tissue, or intra-abdominal infection or dissemination, as evidenced by bloodstream, central nervous system, or another distant site of infection. Patients with invasive disease were categorized according to the primary site from which *Nocardia* spp. were cultured from clinical specimens.

## Microbiological methods

*Nocardia* spp. isolates were recovered from clinical specimens submitted to the clinical microbiology laboratory between 2010 and 2022. All isolates grew aerobically on sheep blood agar and/or specialized growth media (i.e., buffered charcoal yeast extract). Gram stain demonstrated typical gram-positive beaded branching bacilli that were catalase-positive and stained positive using a modified acid-fast stain but negative using a Ziehl-Neelsen stain. Biochemical tests were otherwise not performed. Antibiotic susceptibility testing was performed and interpreted using standard methods outlined in Clinical and Laboratory Standards Institute (CLSI) approved guideline M24S-A3 (12).

Ninety-one isolates were routinely tested using a commercial mass spectrometry system (VITEK MS, bioMérieux, Laval, Quebec, Canada) using the outlined databases in accordance with the manufacturers' instructions. Three isolates recovered prior to 2012 were not available for MALDI-TOF MS analysis (Table 1). Isolates were extracted using ethanol-formic acid protocol according to the manufacturer using the VITEK MS *Mycobacterium*/*Nocardia* Kit IVD (bioMérieux) and subsequently underwent bead-beating according to a previously published protocol prior to proteomics analysis (13). A quantity of 1 µL of the extracted supernatant was placed on the steel target plate, dried, and overlaid with 1 µL of matrix. The target plate was then loaded into the Vitek MS instrument for analysis. Samples were repeated if the result gave low discrimination (<99%) or no identification (ID). VITEK MS results had to give a high confidence (i.e., ≥99.0%) and agree with the results of stains and phenotypic tests results to be reported; all proteomic analyses were performed according to the manufacturer's specifications. The VITEK MS Database V3.2 (released in 2018) was used to interpret results and included some but not all *Nocardia* spp. that were clinically encountered. Because partial sequencing of the 16S rRNA gene and VITEK MS may not provide an accurate species-level identification within *Nocardia* species complexes, data for isolates within the *Nocardia abscessus* complex, *Nocardia cyriacigeorgica* complex, *Nocardia farcinica* complex, *Nocardia nova* complex, *Nocardia paucivorans* complex, *Nocardia otitidiscaviarum* complex, and *Nocardia paucivorans* complex have been presented throughout as the complex rather than individual species within the complex. Antibiotic susceptibility testing was done on 93 isolates using broth microdilution panels (Sensititre Panels, Thermo Fisher Scientific Canada, Mississauga, Ontario) and interpreted according to CLSI supplemental guideline M24S [12]. Antibiotic susceptibility testing data for isolates identified to the genus level (i.e., *Nocardia* spp.) or those with ≤5 isolates represented are not included in Table 2.

## Molecular and phylogenetic analyses

Molecular identification was done by fast polymerase chain reaction/fast cycle sequencing of the 16S rRNA gene (523 bp) with MicroSeq 500 kits and an ABI Prism 3500 XL sequencer (Applied Biosystems, Thermo Fisher Scientific, Foster City, CA, USA) using standard methods previously described (11). A BLAST search against the Smart-Gene Integrated Database Network System (Lausanne, Switzerland) bacterial database indicated the most closely related species (http://www.Smartgene.com), and the overall identity score compared to a well-characterized reference sequence for all isolates was 99.9% with 0–2 mismatches. All bacterial sequence multi-alignments were done against a reference sequence (i.e., listed below for phylogenetics analysis), and the organisms were identified by 16S rRNA gene sequencing interpretation to the genus or species level

**TABLE 1** Performance of VITEK MS compared to the molecular identification of the *Nocardia* species[a,f]

| ID by partial sequencing of the 16S rRNA gene[e] | Reference method ID | | | | |
|---|---|---|---|---|---|
| | No. correct to genus | No. correct to genus and species | No. with discordant species results | No. with no results | Total no. |
| *Nocardia abscessus* complex[b,] | 2 | – | 2 (*N. asiatica*) | – | 2 |
| *Nocardia amamiensis* | | | | 1 | 1 |
| *Nocardia asteroides* | 5 | 5 | | | 5 |
| *Nocardia beijingensis*[b,c] | 3 | 3 | | | 3 |
| Nocardia brasiliensis | 5 | 5 | | | 5 |
| *Nocardia cyriacigeorgica* complex[b,c] | 6 | 5 | 1 (*N. abscessus*) | | 11 |
| | | | | 5 | |
| *Nocardia elegans* | 1 | 1 (*N. farcinica*)[b] | | | 1 |
| *Nocardia endophytica* | | | | 1 | 1 |
| *Nocardia farcinica* complex[b,c] | 21 | 21[c] | | | 21 |
| *Nocardia flavorosea* | | | | 1 | 1 |
| *Nocardia nova* complex[b,c] | 17 | 17 (*N. africans/nova*)[c] | | | 17 |
| *Nocardia otitidiscaviarium* complex | 2 | 2 | | | 2 |
| *Nocardia paucivorans complex*[b,c] | | | | 1 | 1 |
| *Nocardia sputorum* | | | | 1 | 1 |
| *Nocardia transvalensis* complex[b,c] | 12 | 12 | | | 12 |
| *Nocardia* spp.[b,c] | | | | 5 | 5 |
| *Nocardiopsis composita* | | | | 1 | 1 |
| *Nocardiopsis prasina* | | | | 1 | 1 |
| Pseudonocardia antarctica | | | | 1 | 1 |
| Total | 74 (78%) | 70 (77%)[d] | 3 (4%) | 17 (18%) | 91 |

[a]All isolates were analyzed by partial sequencing of the16S rRNA gene sequencing as the reference method. Ninety-one isolates were analyzed by VITEK MS, as three isolates recovered prior to 2012 were no longer available. Proteomics and genomic results were compared to phenotypic results (i.e., Gram stain and phenotypic identification). Closely related species within *Nocardia* species complexes may not be separated by either method.

[b]*Nocardia beijingensis* is part of the *N. abscessus* complex but shown separately, as both methods provided a high-level identification. *N. elegans* is within the *N. nova* complex. *N. farcinica* and *N. kroppenstedii* may not be distinguished by either method. *N. transvalensis* and *N. wallacei* may not be distinguished by either method. *Nocardia* spp. were isolates that 16S sequencing did not provide a species-level identification for.

[c]Accurate genus- and species-level identifications based on the inclusion of the organism in the currently used version of the VITEK MS database. Species not included or with limited species representation in the VITEK MS databases:V3.2 (released in 2018) are highlighted in **bold**; none of these specific genera and/or species were in the database in use at the time of the proteomics analysis and had not been added as of V3.2. VITEK does not distinguish between *N. africans/nova*.

[d]Accurate genus- and species-level identifications based on the total number of clinically relevant *Nocardia* species studied. The total isolates scored for species-level identification by proteomics was decreased in this column if 16S only provided a genus-level identification.

[e]Based on a multi-sequence alignment using a reference strain and a review of the molecular species identification according to the CLSI MM-18 guidelines (14).

[f]Empty cells indicates there are no results for those organisms across the chart.

using published guidelines for interpretive criteria for targeted DNA sequencing analyses published by the CLSI approved guideline MM-18 (14).

A phylogenetic analysis of available *Nocardia* spp. isolates 16S rRNA gene sequences was conducted. The 16S rRNA gene sequence for all clinical isolates (*n* = 117) was aligned with MEGA 11 software (Pennsylvania State University). Twenty reference sequences from culture-type strains were included in the phylogenetic analysis (*Nocardia amamiensis* AB275164, *Nocardia asiatica* AB092566, *Nocardia asteroides* AF430019, *Nocardia beijingensis* AF154129, *Nocardia brasiliensis* AF430038, *Nocardia cyriacigeorgica* AF430027, *Nocardia elegans* AJ854057, *Nocardia endophytica* HM153801, *Nocardia farcinica* AF430033, *Nocardia flavorosea* Z46754, *Nocardia grenadensis* FR729900, *Nocardia kroppenstedtii* DQ157924, *Nocardia nova* AF430028, *Nocardia otitidiscaviarum* AF430067, *Nocardia paucivorans* AF430041, *Nocardia sputorum* LC741024, *Nocardia vulneris* JN705252, *Nocardia wallacei* EU099357, *Nocardiopsis composta* AB368717, *Nocardiopsis prasina* X97884, *Pseudonocardia nitrificans* X55609), all of which are in the *Mycobacteriales* order and *Nocardiaceae* family according to most recent taxonomy available in the LPSN (https://www.bacterio.net/

**TABLE 2** Antimicrobial susceptibility profiles for *Nocardia* species[a,b]

| Species | No. of isolates[c] (n = 80) | Antimicrobials tested (%)[d] | | | | | | | | | |
|---|---|---|---|---|---|---|---|---|---|---|---|
| | | AM/CL | SXT | IMP | CRO | TOB | AMI | CIP | MOX | MIN | LIN |
| *N. abscessus* complex | 13 | 75 | 100 | 67 | 100 | 75 | 89 | 50 | 89 | 88 | 100 |
| *N. cyriacigeorgica* complex | 16 | 88 | 94 | 94 | 100 | 67 | 100 | 50 | 94 | 88 | 94 |
| *N. farcinica* complex | 28 | 81 | 95 | 71 | 67 | 29 | 100 | 67 | 91 | 48 | 100 |
| *N. nova* complex | 16 | 67 | 100 | 83 | 100 | 20 | 100 | 40 | 100 | 40 | 100 |
| *Nocardia transvalensis* complex | 7 | 100 | 100 | 33 | ND | 67 | 100 | 100 | ND | 100 | ND |

[a]Only *Nocardia* species complex data are outlined for those where greater than five isolates were tested. These data do not represent a "true" antibiogram since at least 30 isolates per species/species complex would be needed. Based on the CLSI M24S-A3 MIC breakpoints (12).
[b]ND = antimicrobial not tested.
[c]Several isolates failed to grow for antimicrobial susceptibility testing, including all *Nocardiopsis* sp. (n = 3) and *N. nova* (n = 1).
[d]Antimicrobial abbreviations used: AM/CL = amoxicillin-clavulanate, SXT = trimethoprim-sulfamethoxazole, IMP = imipenem, CRO = ceftriaxone, TOB = tobramycin, AMI = amikacin, CIP = ciprofloxacin, MOX = moxifloxacin, MIN = minocycline, and LIN = linezolid.

). A neighbor-joining tree was inferred with 100 bootstrap replicates, and evolutionary distances were computed using the Jukes-Cantor methods in MEGA 11 (https://www.megasoftware.net/). The tree was manually rooted on a taxonomic outlier (*Pseudonocardia nitrificans*) using FigTree V1.4.4 (https://github.com/rambaut/figtree/releases). Branches corresponding to partitions reproduced in <50% bootstrap replicates were collapsed.

## Data analysis

All data pertaining to individual isolate identification, the 16S rRNA sequence, and the site/source data were initially entered into Microsoft Excel and exported to subsp. version 28V (IBM, Chicago, IL, USA). The cohort data were analyzed using standard descriptive statistics.

## RESULTS

### Patient demographic and clinical characteristics

Ninety-four unique patients with nocardiosis were enrolled. Of these, 91 (96.8%) cases occurred in hospitalized patients, and three cases were ambulatory. Electronic medical records were not available for the latter three patients. Table 3 outlines the demographics, underlying co-morbidities, and overall mortality of the hospitalized nocardiosis cases. All patients were adults >18 years of age with a mean age of 61 ± 17 years. There was no difference between the ages of males and females with invasive infections, but more patients were male (57%) than female (43%). There was no difference in either gender or age distribution according to the species-level identification of *Nocardia* spp. Most patients (n = 89, 95%) had one or more serious underlying co-morbidities, including diabetes, liver, or renal failure, malignancy, transplant recipients, chronic lung disease, rheumatologic diseases, or other inflammatory conditions (Table 1). Underlying chronic lung diseases were frequent (n = 19), and patients had more than one condition, the most common being cystic fibrosis (n = 8, 39%), bronchiectasis (n = 18, 100%), pulmonary fibrosis (n = 6, 33%), chronic obstructive pulmonary disease (n = 7, 39%), and asthma (n = 6, 33%). Active malignancies included acute myelogenous leukemia or chronic lymphocytic leukemia (n = 2, 11%) and various types of lymphoma (n = 2, 11%) and solid organ tumours (n = 13, 72%) (i.e., breast [n = 5, 28%], lung [n = 3, 17%], brain [n = 2, 11%], gastrointestinal [n = 2, 11%], and ureteral [n = 1, 6%]). Transplanted solid organs most commonly were kidney (n = 3, 37%), lung (n = 2, 18%), pancreas (n = 1, 1%), and liver (n = 1, 1%). Another three patients underwent hematopoietic stem cell transplantation (n = 3, 27%). Overall, 1-year mortality was 12.2%, with those having disseminated disease to the CNS most likely to die (n = 4, 67%).

**TABLE 3** Patient characteristics in those with Nocardiosis infections

| Category | N (%) |
|---|---|
| Demographics | N = 94 (100) |
| Mean age, y (SD) | 61 ± 17 |
| Female, mean age, y (SD) | 62 ± 16 |
| Male, mean age, y (SD) | 59 ± 17 |
| Gender | |
| • Male | 52 (57) |
| • Female | 39 (43) |
| Comorbidities | N = 89 (91) |
| • Diabetes | 13 (15) |
| • Liver failure | 3 (3.8) |
| • End-stage renal failure | 7 (8) |
| • Hematologic malignancy | 5 (5.6) |
| • Solid organ tumor | 13 (14.6) |
| • Chronic lung disease | 19 (21) |
| • HSCT recipient[a] | 3 (3.8) |
| • Solid organ transplant | 8 (9) |
| • Rheumatologic diseases[b] | 14 (15.4) |
| • Other inflammatory conditions[c] | 18 (19.8) |
| 1-Year mortality | N = 12 (12.2%) |
| 1-Year mortality with CNS disease | N = 4/6 (67%) |

[a]HSCT = hematopoetic stem cell transplant.
[b]Rheumatologic diseases (n = 14): rheumatoid arthritis (1), systemic lupus erythematosus (3), osteoarthritis (2), sarcoidosis (2), psoriasis (2), anti-phospholipid syndrome (APLAs) (1), polymyalgia rheumatica (2), and Sjogren's syndrome (1).
[c]Other inflammatory conditions (n = 18): abscesses (n = 5, 28), SSTI (N = 3, 17), inflammatory bowel disease (n = 3, 27), membranoproliferative glomerulonephritis (n = 2,11), and one each of pancreatitis, hepatitis, gastrointestinal reflux diseases (GERD), autoimmune hemolytic anemia, and allergic bronchopulmonary aspergillosis.

## Performance of MALDI-TOF MS for *Nocardia* spp. identification

Table 2 compares the performance of MALDI-TOF MS for identification of 91 of the *Nocardia* spp. isolates. VITEK MS provided a genus-level identification for most isolates (78%) and a species-level identification for 77%. VITEK MS does not distinguish between *N. africanus/nova*, so these results were scored as accurate species-level identifications. There were three discordant results for species as outlined in Table 2; *N. elegans* is within the *N. farcinica* complex. A total of 17 isolates (18%) could not be identified by MALTI-TOF MS, and four (25%) of these were not included in the VITEK MS V3.2 database (i.e., highlighted in 'bold' in Table 2, including *N. amamiensis*, *N. elegans*, *N. endophytica*, *N. paucivorans* complex, and *N. sputorum*). In comparison, 16S sequencing provided an accurate species/species complex identification for 93.6% of all 94 isolates; only a genus-level identification was provided for six isolates.

## Distribution of *Nocardia* spp. according to phylogeny and site of infection

Most infections were either due to primary pulmonary (n = 48, 53%) or SSTIs (n = 23, 27%). SSTIs included cellulitis and abscesses (i.e., scalp, forearms, legs, and ankle), erythematous plaque-like lesions, and one case of breast implant infection. BSIs (n = 7, 8%), CNS infections (n = 6, 7%), septic arthritis (n = 3, 3%), and intra-abdominal infections (IAI) (n = 2, 2%) occurred less frequently. BSIs mainly originated from pulmonary infection (80%), with one case having prosthetic mitral valve endocarditis due to *N. composta*. All CNS infections originated from primary pulmonary infections, but one patient also had skin lesions. Septic arthritis occurred in one native elbow joint and two cases of prosthetic joint infection (i.e., one knee and one hip). Another five patients had nocardiosis of another unique site, including the olecranon bursitis (n = 2), central catheter site infection (n = 2), and endophthalmitis (n = 1).

Fig. 1 shows the distribution of all *Nocardia* species recovered by culture from various clinical specimen sites/sources. Although a wide variety of different *Nocardia* spp. were isolated, over half of infections (56/94 [60%]) were caused by three species complexes, namely, *Nocardia farcinica* (*n* = 21, 37.5%), *Nocardia nova* (*n* = 19, 20%), and *Nocardia cyriacigeorgica* (*n* = 16, 17%). These three species complexes primarily caused pulmonary infections (33/48, 69%), SSTIs (14/23, 61%), and BSIs (5/7, 71%). *N. brasiliensis* only caused superficial SSTIs (7/23, 30%). *N. asteroides* primarily caused pulmonary infection (4/5, 80%). The *N. farcinica* complex also caused most CNS infections (4/6 [67%]) and all intra-abdominal infections. Infections were otherwise caused by a wide variety of different *Nocardia* spp. (Fig. 1 and 2).

Fig. 2 shows the phylogenetic correlation of *Nocardia* spp. 16S rRNA gene sequences (*n* = 117) according to site of infection. Sixteen patients with pulmonary infections had the same *Nocardia* spp. isolated on multiple occasions over the course of their follow-up, often separated by 2 or more years apart. Several correlations were observed between the site of infection and *Nocardia* spp. Most 16S isolate sequences were from pulmonary infections (*n* = 71, 61%), and the *N. transvalensis* complex (*n* = 17, 24%) was predominantly isolated, followed by similar proportions of cases due to the *N. nova* complex (*n* = 12, 17%), *N. cyriacigeorgica* complex (*n* = 11, 16%), *N. farcinica* complex (*n* = 10, 14%), and *N. abscessus* complex (i.e., *N. asiatica* and *N. beijingensis* [*n* = 9, 13%]). The remaining 12 pulmonary isolates (16%) were distributed among eight different *Nocardia* species. SSTIs (*n* = 23, 19.6%) were caused by the *N. farcinica* complex (*n* = 9, 39%), *N. brasiliensis* (*n* = 7, 30%), and *N. vulneris* (*n* = 7, 30%), which clustered closely together, *N. cyriacigeorgica* complex (*n* = 4, 17%), and one isolate each of *N. beijingensis*, *N. nova* complex, and *N. otitidiscaviarum* complex. BSIs (*n* = 7, 6%) were caused by the *N. nova* complex (*n* = 3, 43%) and several different *Nocardia* spp. (i.e., *N. farcinica* complex, *N. cyriacigeorgica* complex, *N. asteroides*, and *N. composta*) (*n* = 4, 57%). CNS infections (*n* = 6, 4%) were mainly caused by the *N. farcinica* complex (*n* = 4, 67%), while the two other cases were

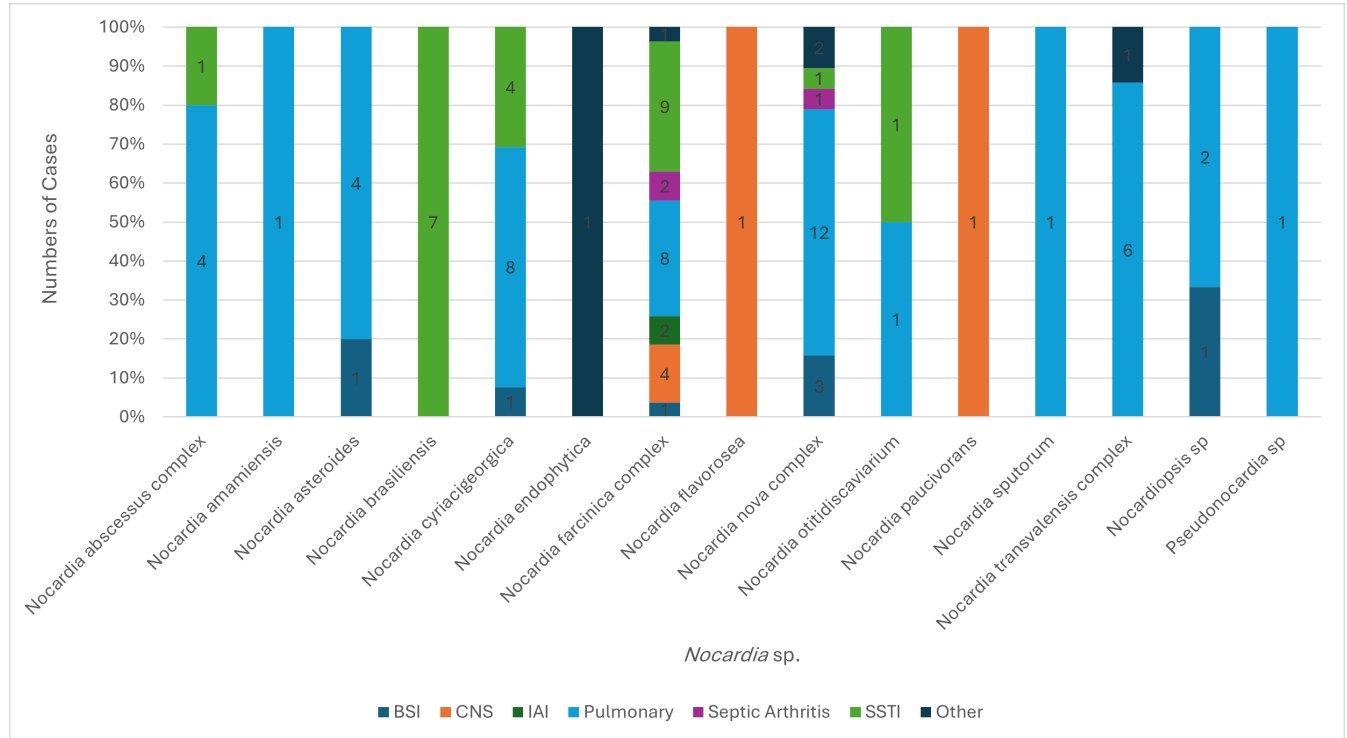

**FIG 1** Distribution of *Nocardia* species causing various types of nocardiosis infection. BSI = bloodstream infection, CNS = central nervous system infection, IAI = intra-abdominal infection, and SSTI = skin and soft tissue infection. Two additional species also represent species complexes, including *N. cyriciageorgica* and *N. paucivorans*.

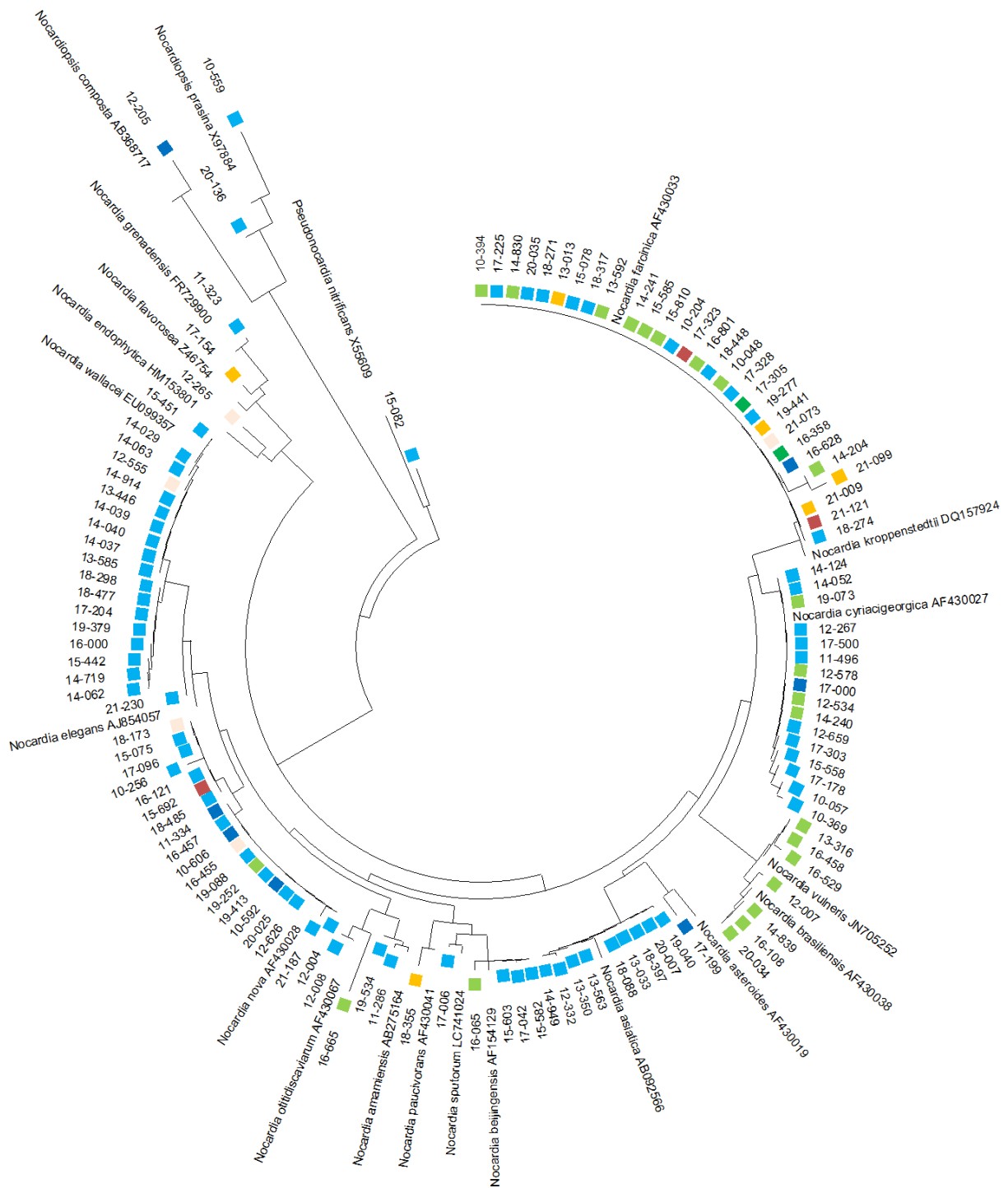

**FIG 2** Phylogenetic tree of the *Nocardia* isolate 16S sequences.

caused by *N. flavorosea* and *N. paucivorans* complex. Septic arthritis of an elbow was due to the *N. nova* complex, while the *N. farcinica* complex caused the two cases of PJI. Both cases of IAI were caused by the *N. farcinica* complex. Two *N. nova* complexes and one each of *N. endophytica*, *N. farcinica* complex, and *N. transvalensis* complex caused five infections at other sites.

## Antimicrobial susceptibility

Susceptibility testing was performed for 93 (99%) isolates, but not all drugs were tested against all isolates based on the site of recovery. Table 3 shows the antibiotic profiles for the most common *Nocardia* species complexes against most antimicrobial agents used to treat nocardiosis. Most or all *N. cyriacigeorgica* complex and *N. nova* complex isolates were susceptible to trimethoprim-sulfamethoxazole (SXT), imipenem (IMP), ceftriaxone (CRO), amikacin (AMI), moxifloxacin (MOX), and linezolid (LIN), but they had increased resistance to other tested drugs. The *N. nova* complex isolates were, however, more frequently resistant to IMP (83% susceptible) compared to the *N. cyriacigeorgica* complex (94% susceptible). The *N. farcinica* complex had a unique antibiogram, with only SXT, AMI, and LIN drugs testing as >95% susceptible, while resistance was shown for other agents, including CRO (67%). The *N. vulneris* isolates also had increased resistance to CRO (67%). Resistance across major drug classes also varied by species. Although most *Nocardia* spp. were susceptible to AMI (except for *N. asiatica*), tobramycin (TOB) susceptibility widely varied. Among the quinolones, MOX was more active than ciprofloxacin against *Nocardia* spp., where both drugs were tested. A small number of other *Nocardia* spp. (≤5 isolates for individual species/species complexes), including *N. amamiensis*, *N. elegans*, *N. flavorosea*, *N. otitidiscaviarum* complex, *N. paucivorans* complex, as well as *Pseudonocardia* spp., were susceptible to all the drugs tested. *N. sputorum* was resistant to IMI and CRO but otherwise susceptible (Table 3).

## DISCUSSION

Our unique multi-year population-based study of *Nocardia* spp. recovered from a regional cohort of patients with nocardiosis is one of the largest comparisons of disease presentation with the isolates' microbiological characteristics in which a species-level identification was done through both 16S rRNA gene sequencing and MALDI-TOF MS. There has been a scarcity of population-based studies of nocardiosis, and few studies correlated specific *Nocardia* spp. phylogeny using a specific gene target (i.e., 16S rRNA, *sec*1, and *sod*A) to clinical disease manifestations (15–17). Although a wide variety of different *Nocardia* spp. caused nocardiosis in our region, more than half of these cases were caused by three species complexes, including *N. cyriacigeorgica*, *N. farcinica*, and *N. nova*. Clinical and phylogenetic analyses showed specific correlations of disease manifestations to individual *Nocardia* spp. The above triad of *Nocardia* species caused most pulmonary, SSTIs, and invasive infections. *N. farcinica* complex, however, appears to cause most CNS infections in our region and has increased antibiotic resistance to IMP and CRO, which are commonly used to empirically treat brain infections (4). Laboratories should either perform antibiotic susceptibility testing or refer invasive *Nocardia* spp. isolates to a reference laboratory given the unique and evolving drug profiles for commonly used agents. A regional antibiogram of the most isolated *Nocardia* spp. would also better direct initial empiric therapy.

Our work enhances understanding of regional differences in clinical and epidemiological aspects of nocardiosis diagnosis due to the emerging niches of recently described *Nocardia* species and their different susceptibility patterns. Although *N. asteroides* is reportedly the most isolated *Nocardia* spp. to date (1), this was not found in our region or in the recent reports from other distinct geographic parts of the world that have used advanced methods to characterize clinical *Nocardia* spp. isolates over a longitudinal period. Several recent reports described below highlight the clinical importance of several species' complexes (i.e., *N. nova*, *N. cyriacigeorgica*, *N. farcinica*, and *N. abscessus*) in predominantly causing pulmonary infections with dissemination to the bloodstream, CNS, and other distant locations. These reports also show that *N. brasiliensis* and *N. otitidiscaviarum* complex mainly cause SSTIs. Overall, *Nocardia* spp. have similar antibiotic susceptibility profiles as those described earlier (1, 2), except for more recently described species where limited data are available. Few studies have been reported from North America and Europe. Gupta and colleagues (18) reviewed 268 patients with invasive *Nocardia* infections in three non-contiguous geographic areas

in the USA during 2011–2018: 48.2% from Minnesota, 32.4% from Arizona, and 19.4% from Florida. Although the predominant species/species complexes isolated were like ours, there were regional differences in the most common isolates: the *N. nova* complex was common in Minnesota (33.4%), *N. cyriacigeorgica* complex in Arizona (41.4%), and *N. brasiliensis* in Florida (17.3%). Pulmonary infection occurred in more than two-thirds of the patients, regardless of the immune status, but CNS involvement occurred much less often. Antibiotic susceptibility patterns for all major *Nocardia* spp. were similar with no variation across the three locations. Valdezate and colleagues (19) in Europe (Spain) performed a multi-year study of 1,119 Nocardia strains and showed a similar diversity to our study, with most isolates belonging to the *N. cyriacigeorgica* complex > *N. nova* complex > *N. abscessus* complex, or *N. farcinica* complex. Most patients (~86%) had pulmonary infections, and *N. farcinica* complex strains caused one-half of the CNS infections. Linezolid and amikacin were the most frequently active agents, with several species having variable resistance rates to other commonly reported agents.

Three large multi-year studies have also been reported from Southeast Asia or Australia and New Zealand. Yeoh and colleagues (10) conducted a retrospective review of 414 *Nocardia* spp. isolates representing 27 species/species complexes referred to the Mycobacterial Reference Laboratory in Victoria, Australia between 2009 and 2019. Most patients had pulmonary infection due to either the *N. nova* complex or the *N. cyriacigeorgica* complex. The *N. farcinica* complex and the *N. paucivorans* complex were, however, more commonly recovered from sterile sites. *N. brasiliensis* and *N. otitidiscaviarum* complex were also mostly isolated from SSTIs. Linezolid and amikacin had the highest proportion of susceptible isolates, while low susceptibility rates were detected in the *N. farcinica* complex isolates for ceftriaxone (59%) and imipenem (41%). O'Brien and colleagues (20) identified 960 isolates in Australia (2005–2022) through the Australian Passive Antimicrobial Resistance Surveillance program, giving an annual incidence of 3.03 per 100,000 population. The four most common species/species complexes were *N. brasiliensis*, *N. nova*, *N. cyriacigeorgica*, and *N. farcinica*. All isolates had high rates of susceptibility to linezolid (100%) and trimethoprim sulfamethoxazole (98%). Amikacin (90%) susceptibility for all isolates, but the *N. transvalensis* complex, was more active than carbapenems or third-generation cephalosporins. Susceptibility to other oral agents demonstrated significant interspecies variation. McKinney and colleagues (21) identified 383 isolates in New Zealand over a 20-year period, and, again, the *N. nova* complex (226, 59%), *N. cyriacigeorgica* complex (42, 11%), and *N. farcinica* complex (41, 11%) were most recovered, but a diverse number of other species/species complex were represented (51, 13%). *Nocardia* spp. isolates mainly caused pulmonary infections (244, 64%). One-third of the patients had SSTIs (104, 27%), and all *N. brasiliensis* isolates were recovered from this group. Almost all isolates were susceptible to amikacin, linezolid, and trimethoprim-sulfamethoxazole, but 35.6 and 77% were resistant to clarithromycin and quinolones, respectively. Wang and colleagues (22) in China studied 791 *Nocardia* isolates. The most common species/species complex included *N. farcinica* (29.1%, 230/791), followed by *N. cyriacigeorgica* (25.3%, 200/791), *N. brasiliensis* (11.8%, 93/791), and *N. otitidiscaviarum* (7.8%, 62/791). The first two species complexes caused most pulmonary infections, while the latter two species/species complexes caused SSTIs and extra-pulmonary infections. Although susceptibility varied by *Nocardia* spp., linezolid (99.5%), amikacin (96%), and trimethoprim-sulfamethoxazole (92.9%) were the most active agents, with imipenem (64.7%) being less susceptible across all species.

Our study also highlights the improvement in providing a species-level identification of *Nocardia* spp. by implementation of advanced proteomics and genomics methods. Prior to the widespread use of MALDI-TOF MS systems in clinical microbiology, laboratories relied on a combination of morphologic, phenotypic, and genomic analyses for identification of this complex group of organisms. Ours is one of the few studies to compare VITEK MS performance with partial sequencing of the 16S rRNA gene. Although a partial sequence of this gene target may also not be able to accurately distinguish species within *Nocardia* species complexes, it nevertheless remains the "gold standard

used by many clinical laboratories" (14). Although VITEK MS using an appropriate extraction profile provided a reliable species-level identification in two-thirds of our isolates, partial or complete sequencing of the 16S rRNA gene provided an accurate species/species complex identification for 93.6%. Of the 94 *Nocardia* isolates included in our study, 16S sequencing failed to provide an accurate species or species complex identification for six. While VITEK MS failed to provide an accurate species or species complex identification for 23% of isolates, it provided discordant species results for another 4% and no identification results for another 18%. Based on our VITEK MS study, 77% of *Nocardia* isolates were identified to the species level. However, these data also demonstrate that neither method may provide an accurate Nocardia species or species complex identification in between 6 and 23% of isolates, and in ~20% of the cases, MALDI-TOF MS provided an inaccurate result of no result.

To date, other performance data for the accurate identification of *Nocardia* spp. using either the VITEK MS or Bruker MS systems are limited. Previously reported VITEK MS studies of the performance of MALDI-TOF MS using the V3.2 database for *Nocardia* identification have shown similar genus- and species-level identification results. Hodille and colleagues (23) compared 16S sequencing to VITEK MS (V3.2) identification using 134 *Nocardia* isolates and found that MALDI-TOF MS gave an interpretable result for most (81.3%) isolates with an overall agreement with the reference method of 78.4%. Performance was improved up to 94% by excluding *Nocardia* spp. not contained within the instrument's database (23, 24). Toyokawa and colleagues (25) also characterized 153 clinical *Nocardia* spp. isolates from Japan by both MALDI-TOF MS (Bruker Daltonics) and 16S rRNA gene sequencing. *N. farcinica* (25%) was the most common species, followed by *N. cyriacigeorgica* (18%), *N. brasiliensis* (9%), *N. nova* (8%), and *N. otitidiscaviarum* (7%). MALDI-TOF MS with the use of a supplemental *Nocardia* library correctly identified 97.3% ($n = 146$) to the species/complex level and 1.3% ($n = 2$) to the genus level. MALDI-TOF MS, therefore, provides a good initial species-level identification of commonly isolated *Nocardia* spp., but the Bruker system may have better performance compared to the VITEK MS, albeit based on the limited data published to date due to the breadth of the supplemental *Nocardia* database. However, no head-to-head studies have been reported comparing the performance of VITEK MS and Bruker MS to molecular identification of *Nocardia* species/species complexes.

Since this was a multi-year study, some limitations must be considered in this microbiological cohort. Due to the limitations of phenotypic methods and VITEK MS databases, some cases may have been misidentified or missed (i.e., a wrong identification or no result occurred for ~25% of the cases). However, this is less likely in our study since all suspected *Nocardia* isolates were also analyzed by 16S sequencing (26). However, improved molecular species-level separation of some *Nocardia* spp. within important complexes would have been achieved by interrogation of a longer portion of the 16S gene (i.e., up to ~1,060 bps or even the entire gene) and/or analysis of additional gene targets, such as *rpo*B, *sod*A, or *sec*1 (14, 16, 17, 27). Whole-gene sequencing may also be used in future studies to definitively separate newly emerging rarely isolated *Nocardia* spp., although recent evaluation of WGS compared to 16S rRNA gene sequencing provided equivalent results (1, 28).

In summary, this work further advances the understanding of the unique clinical and epidemiological roles of *Nocardia* spp. with a widening spectrum of presentations and pathogenesis. From a laboratory perspective, a species-level identification of *Nocardia* spp. is necessary for isolates causing invasive disease due to the unique distribution of species causing major types of infection demonstrated by this and other published studies. In addition, a major species/species complex identification allows more accurate selection of empiric treatment for nocardiosis based on known antibiotic profiles. More extensive population-based studies, however, are required to further link microbiological and clinical characteristics with outcomes to fully understand the pathogenesis of this complex group of organisms.

## Conclusions

A wide variety of different *Nocardia* spp. caused nocardiosis in our region during our multi-year population-based study, but more than half of these cases were caused by *N. cyriacigeorgica*, *N. farcinica*, and *N. nova*. Because molecular methods are continuously defining taxonomic changes within Nocardiae, epidemiologic studies should analyze cases by species to improve understanding of the clinical and epidemiological niches of this complex group of organisms. Empiric treatment of nocardiosis with one or more active agents requires access to regional antibiotic profile data.

## ACKNOWLEDGMENTS

We thank the Alberta Precision Laboratory—Calgary Zone Microbiology staff for their help with sample processing and isolate analysis.

This study was unsupported.

D.L.C.: conceptualization, data curation, methodology, project administration, writing—original draft, and writing—review and editing. G.P.: data curation, phylogenetic analysis and tree construction, writing—original draft, and writing—review and editing. A.U.T.: data curation, methodology, and writing—review and editing. C.N.: data analysis, methodology, and writing—review and editing.

## AUTHOR AFFILIATIONS

[1]Section of Infectious Diseases, Department of Medicine, Cummings School of Medicine, University of Calgary, Alberta, Canada

[2]Department of Pathology and Laboratory Medicine, Cummings School of Medicine, University of Calgary, Alberta, Canada

[3]Division of Microbiology, Alberta Precision Laboratories, Alberta Health Services, Calgary, Alberta, Canada

[4]Department of Community Health Sciences, Cummings School of Medicine, University of Calgary, Calgary, Alberta, Canada

## AUTHOR ORCIDs

Deirdre Church  http://orcid.org/0000-0002-8306-8099

## AUTHOR CONTRIBUTIONS

Deirdre Church, Conceptualization, Data curation, Formal analysis, Investigation, Methodology, Project administration, Supervision, Writing – original draft, Writing – review and editing | Gisele Peirano, Data curation, Formal analysis, Methodology, Writing – original draft, Writing – review and editing | Alejandra Ugarte-Torres, Conceptualization, Data curation, Methodology, Writing – review and editing | Christopher Naugler, Conceptualization, Formal analysis, Methodology, Writing – original draft, Writing – review and editing

## DATA AVAILABILITY

The data that support the findings of this study are available from Alberta Health Services (AHS), Alberta Precision Laboratories (APL) (formerly CLS), but restrictions apply to the availability of these data, which were used under the ethics agreement for the current study and so are not publicly available. Data are, however, available from the author upon reasonable request and with permission of the AHS/APL.

All sequences were submitted to GenBank under accession numbers MH190229 to MH190303, OR641424 to OR641466, OR805488, OR805489, PQ153237, and PQ153238.

## ETHICS APPROVAL

A waiver of consent requirement was obtained, and the study was reviewed by the Conjoint Health Research Ethics Board (REB) and approved under certificate number REB22-1280.

## ADDITIONAL FILES

The following material is available online.

### Open Peer Review

**PEER REVIEW HISTORY (review-history.pdf).** An accounting of the reviewer comments and feedback.

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
