## [Reviewer comments · Microbiology Spectrum]

Microbiology Spectrum

Population-based Microbiological Characterization of *Nocardia* Strains Causing Invasive Infections during a Multiyear Period in a Large Canadian Health Care Region

Deirdre Church, Gisele Peirano, Alejandra Ugarte-Torres, and Christopher Naugler

Corresponding Author(s): Deirdre Church, University of Calgary Cumming School of Medicine

Review Timeline:

Submission Date:	March 27, 2025
Editorial Decision:	April 27, 2025
Revision Received:	May 14, 2025
Accepted:	May 15, 2025

Editor: Karen Carroll

Reviewer(s): Disclosure of reviewer identity is with reference to reviewer comments included in decision letter(s). The following individuals involved in review of your submission have agreed to reveal their identity: Benjamin Douglas Moser (Reviewer #1)

Transaction Report:

DOI: <https://doi.org/10.1128/spectrum.00914-25>

Re: Spectrum00914-25 (**Population-based Microbiological Characterization of *Nocardia* Strains Causing Invasive Infections during a Multiyear Period in a Large Canadian Health Care Region**)

Dear Dr. Deirdre L. Church:

Thank you for the privilege of reviewing your work. Below you will find my comments, instructions from the Spectrum editorial office, and the reviewer comments.

Thank you for submitting your manuscript to Microbiology Spectrum. While I agree with reviewer 1 that the authors are to be commended for the detailed evaluation of *Nocardia* infections in the MALDI-TOF MS era, I share the multiple major concerns of Reviewer #2. My major criticism is that neither MALDI-TOF MS nor sequencing of the first 500 base pairs of the 16S rRNA gene are adequate in distinguishing among the major *Nocardia* complexes. For example, in Table 1, the authors note the discrepancy between *Nocardia abscessus* and *Nocardia asiatica*. In reality, these two species are part of the same complex and cannot reliably be distinguished by the methods used in this paper. Likewise, *N. wallacei* which is in the *N. transvalensis* complex is more common than *N. transvalensis sensu strictu*. Hence, identification to exact species level is overstated. When referring to *N. transvalensis*, *N. nova* and *N. abscessus*, the word "complex" should be used to describe them. Others have assessed non-16S rRNA genes such as *secA*, and *hsp* as more reliable targets for species level identification. If you are willing to make these major revisions (which may involve additional data analysis) including discussion of the limitations of the methods, then I am willing to consider the revised work for possible publication.

Revision Guidelines

Sincerely,
Karen Carroll

Reviewer #1 (Comments for the Author):

Excellent review for a tricky diagnosis. Glad to better identification and abilities to treat in a more nuanced and strain specific way.

Reviewer #2 (Comments for the Author):

In the manuscript "Population-based Microbial Characterization of Nocardia Strains Causing Invasive Infections during a Multiyear Period in a Large Canadian Health Care Region," Church et al. make correlations between nocardiosis, organism genus and species, and antimicrobial susceptibility results. The manuscript has some inconsistencies that need to be addressed, with some recommendations for the methods and discussion to clarify claims for the reader.

Major comments:

1. Inconsistencies:

- a. Manuscript swaps between 2012-2019 and 2014-2022 as the years this study was conducted.
- b. Numbers/percentages in lines 220-221 (94 total) don't match Table 2 (91 total).
- c. Complex vs. species vs. species/species names used: e.g. Fig 1 N. transvalensis complex, Line 230 N. wallacei, and Table 2 N. transvalensis/wallacei.
- d. Numbers in Table 1 do not match numbers in Table 1 footnotes.
2. Line 81: Nocardia asteroides is suggested to be the most reported species isolated from human specimens, but this is not mentioned in this study or any highlighted in the discussion.
3. Line 118: Ziehl-Neelsen is an acid-fast stain and will not identify only modified-acid fast bacilli.
4. Please provide additional details to methods:
 - a. AST details - these can be brief (e.g. microbroth dilution, media type) as this isn't routinely performed by non-reference laboratories.
 - b. MALDI TOF MS - can you discuss this as claimed vs. unclaimed organisms - unclear what you mean by "non-research database" (line 123). Add kit used for extraction.
 - c. Line VITEK MS does not give results in a log score.
 - d. Line 140: What were the reference sequences used for alignment?
5. Clarify results/discussion:
 - a. Add n/total, % for comparisons to add context (e.g. N. brasiliensis only caused superficial SSTIs)
 - b. Line 230: Trying to communicate that
 - c. Table 2. Simplify/clarify footnotes. Move "No. (%)" from columns to row labeled TOTAL.
 - d. Line 342: be more specific - "without use of the research database"
6. Table 3 Antibigram data. An Antibigram ideally will include > 30 isolates to represent a range of results. Even with only 10 isolates represented, this data is usually presented with some type of disclaimer. I would recommend deleting all isolates only represented once, maybe even those with <5. Can you group these in with "Nocardia sp."?
 - a. Add confidence intervals to communicate the error/variability in representing such a small number of isolates.
 - b. What is "Nocardia sp." - does this mean sequencing didn't get a species-level identification?
 - c. Lines 258-260 - delete - appears 100% susceptible to all drugs because there was only one isolate of each represented.
7. Lines 283-335 need to be summarized as to highlight similarities and/or differences compared to this study's data. This is hard to follow
8. I would like to see further discussion of the results and perceived implications of the differences in identification capabilities between sequencing (reference method) and MALDI-TOF MS - as nearly 20% had no ID, and there were additional with no species.
9. Expand on MALDI TOF MS differences - Line 354: claim that Bruker performs better due to breadth of the database. How do these compare?
10. Line 360: Improvements of sequencing discussed, but the data and limitations were not forthcoming or clear. What is the gap that needs improving?

Minor comments:

1. Unclear what line 230-233 is trying to say.
2. Define abbreviations before using.
3. Disease state nocardiosis doesn't need to be capitalized.

In the manuscript “Population-based Microbial Characterization of *Nocardia* Strains Causing Invasive Infections during a Multiyear Period in a Large Canadian Health Care Region,” Church et al. make correlations between nocardiosis disease presentation, organism genus and species, and antimicrobial susceptibility results. The manuscript has some inconsistencies that need to be addressed, with some recommendations for the methods and discussion as well.

Major comments:

1. Inconsistencies:
 - a. Manuscript swaps between 2012-2019 and 2014-2022 as the years this study was conducted.
 - b. Numbers/percentages in lines 220—221 (94 total) don't match Table 2 (91 total).
 - c. Complex vs. species vs. species/species names used: e.g. Fig 1 *N. transvalensis* complex, Line 230 *N. wallacei*, and Table 2 *N. transvalensis/wallacei*.
 - d. Numbers in Table 1 do not match numbers in Table 1 footnotes.
2. Line 81: *Nocardia asteroides* is suggested to be the most reported species isolated from human specimens, but this is not mentioned in this study or any highlighted in the discussion.
3. Line 118: Ziehl-Neelsen is an acid-fast stain and will not identify only modified-acid fast bacilli.
4. Please provide additional details to methods:
 - a. AST details – these can be brief (e.g. microbroth dilution, media type) as this isn't routinely performed by non-reference laboratories.
 - b. MALDI TOF MS – can you discuss this as claimed vs. unclaimed organisms – unclear what you mean by “non-research database” (line 123). Add kit used for extraction.
 - c. Line VITEK MS does not give results in a log score.
 - d. Line 140: What were the reference sequences used for alignment?
5. Clarify results/discussion:
 - a. Add n/total, % for comparisons to add context (e.g. *N. brasiliensis* only caused superficial SSTIs)
 - b. Line 230: Trying to communicate that
 - c. Table 2. Simplify/clarify footnotes. Move “No. (%)” from columns to row labeled TOTAL.
 - d. Line 342: be more specific - “without use of the research database”
6. Table 3 Antibiogram data. An Antibiogram ideally will include > 30 isolates to represent a range of results. Even with only 10 isolates represented, this data is

usually presented with some type of disclaimer. I would recommend deleting all isolates only represented once, maybe even those with <5. Can you group these in with "Nocardia sp."?

- a. Add confidence intervals to communicate the error/variability in representing such a small number of isolates.
 - b. What is "Nocardia sp." – does this mean sequencing didn't get a species-level identification?
 - c. Lines 258-260 – delete – appears 100% susceptible to all drugs because there was only one isolate of each represented.
7. Lines 283-335 need to be summarized as to highlight similarities and/or differences compared to this study's data. This is hard to follow
 8. I would like to see further discussion of the results and perceived implications of the differences in identification capabilities between sequencing (reference method) and MALDI-TOF MS – as nearly 20% had no ID, and there were additional with no species.
 9. Expand on MALDI TOF MS differences – Line 354: claim that Bruker performs better due to breadth of the database. How do these compare?
 10. Line 360: Improvements of sequencing discussed, but the data and limitations were not forthcoming or clear. What is the gap that needs improving?

Minor comments:

1. Unclear what line 230-233 is trying to say.
2. Define abbreviations before using.
3. Disease state nocardiosis doesn't need to be capitalized.

Editorial/Reviewers Comments	Response	Changes to the Manuscript	Comments
Neither MALDI-TOF MS or 16S sequencing may distinguish among the major Nocardia complexes	We checked the reliability of all 16S sequences for quality and accuracy of species-level identification. A reliable species ID was obtained for all but 6/94 isolates using 16S sequencing (93.6%). However, we have made major modifications throughout the manuscript to address this overlying concern.	Lines 138-144: Methods have been revised to make it clear that either methods may not provide a species/species complex ID. All species-level identifications have been changed to Nocardia species complexes where appropriate (i.e., N. abscessus complex, N. cyriaciageorgica complex, N. farcinica complex, N. paucivorans complex, N. otitidiscaviarum complex and N. transvalensis complex.	
Reviewer #2			
Ensure consistency in the years of study	The first isolates were enrolled in 2010 and the last in 2022.	Between 2010-2022 has been used throughout the manuscript	
Table 2: Ensure consistency in the numbers/percentages with the text lines 220-221	A total of 94 isolates were enrolled but only 91 were analyzed by MALDI-TOF MS.	The numbers/ Percentages in Table 2 are now consistent with the text. The discrepancy between the total number of isolates and those analyzed by MALDI-TOF MS is outlined in the text (Methods and Results).	
Ensure consistency in the complex vs. species/species names used throughout		All species vs. species-complex names have been reviewed in the text and illustrations and made consistent. As outlined above species complex names are now used throughout where appropriate.	
Line 81: N. asteroides is referenced as being the most reported species but this is not mentioned in the study or highlighted in the Discussion.		This sentence has been modified to indicate (lines 82-83) “but few cases have been caused by this species in our region.” N. asteroides only caused 5 cases during the study – this is outlined in the results (line 258). A sentence was added	

		to the Discussion (Lines 305-308).	
Line 118: Modify this sentence		Line 119-121: sentence has been modified to indicate the right staining pattern.	
Add additional details to Methods for AST		Lines 144-148 outlined the AST Methods	
Separate MALDI-TOF MS claimed vs unclaimed (not in database) organisms and provide the extraction kit	Only 4/17 species that were not identified by VITEK MS were not included in the V3.2 database.	The 4 species not included in the VITEK MS V3.2 database are highlighted in 'bold' in Table 2 and line 220-222 outlines this in the text. The kit information has been included in lines 129-30.	
VITEK MS doesn't give results in log score.		Line 134: this has been corrected	
Line 140: What were the reference sequences used for alignment?	The reference sequences were previously listed under Methods – Phylogenetic analysis.	All of the details about molecular analysis have been moved into a new Methods section called Molecular and Phylogenetic analysis. Lines 140-160 outline that the reference sequences are listed below under phylogenetic analyses.	
Add n/total, % for N. brasiliensis only caused superficial SSTIs. Clarify Line 230.		Line 241: This has been added for context.	
Table 2: Simplify as outlined		Table 2: % have been moved to the TOTALS. All footnotes were reviewed and cited in the text. All species complexes used as proper taxonomy where appropriate.	
Line 342: Be more specific	Although VITKE MS has a research database called Suramis, this was not used during the study. All MALDI-TOF MS results were interpreted against the v3.2 instrument database.	The "without use of the research database" has been removed.	

Table 3 Antibiogram data – condense to only include the largest groups of organisms as not a ‘true’ antibiogram.	To date, there are few regional studies that have accrued enough isolates within rarely isolated Nocardia spp. to produce a ‘true’ antibiogram. However, compilation of specific antibiotic profiles (<30 organisms per species or species complex) is clinically important on a regional basis.	Table 3 has been modified to only include the most prevalent Nocardia species complexes. Table 3 Title has also been changed to call these antibiotic profiles. The Results text has been altered accordingly to align with the new Table 3.	Confidence intervals were not included as all results were reported by the laboratory as an interpretation, not as an actual MIC result.
What are Nocardia spp.?	Nocardia spp. included isolates that 16S didn’t get a species/level ID.	This has been clarified in the text (Lines 222-223).	
Lines 258-260. Delete this sentence as only a single isolate for each species.		Lines 279-283: This sentence has been modified to indicate it represents a small number of isolates.	
Lines 283-335 Modify the Discussion	The Discussion has undergone major revision for clarification so that major differences/similarities between ours and other studies are highlighted.	Lines 303-360: A major revision of the Discussion was done.	
Discussion: Enhance discussion of the differences between MALDI-TOF MS and 16S for identification of Nocardia spp.		See Lines 361-377: An expanded discussion is now included of the differences between MALDI-TOF MS and 16S sequencing for Nocardia species/species complex identification.	
Line 354: Expand the discussion about Bruker		See Lines 384-395: This section has been expanded to include a better discussion of Bruker.	
Line 360: Expand on sequencing improvements		See Lines 401-406: This highlights the limitations of partial 16S sequencing and the potential use of WGS.	
Lines 230-233: Clarify this		Lines 248-250: This sentence has been	

sentence		modified.	
Define abbreviations before using		All abbreviations have been reviewed and defined when first used	
Nocardiosis doesn't need to be capitalized		This has been corrected throughout.	

Re: Spectrum00914-25R1 (**Population-based Microbiological Characterization of *Nocardia* Strains Causing Invasive Infections during a Multiyear Period in a Large Canadian Health Care Region**)

Dear Dr. Deirdre L. Church:

Thank you for addressing the reviewers' comments and for the major revisions to the manuscript. There are several typographical and punctuation errors that can be corrected during the proof stage. Overall, I think the manuscript is much improved and I am moving it forward for publication.

Your manuscript has been accepted, and I am forwarding it to the ASM production staff for publication. Your paper will first be checked to make sure all elements meet the technical requirements. ASM staff will contact you if anything needs to be revised before copyediting and production can begin. Otherwise, you will be notified when your proofs are ready to be viewed.

Sincerely,
Karen Carroll
Editor
Microbiology Spectrum